# Homo/Hetero-Dimers of Aromatic Bisabolane Sesquiterpenoids with Neuroprotective Activity from the Fungus *Aspergillus* *versicolor* A18 from South China Sea

**DOI:** 10.3390/md20050322

**Published:** 2022-05-13

**Authors:** Han-Zhuang Weng, Jun-Yu Zhu, Fang-Yu Yuan, Zhuo-Ya Tang, Xiao-Qing Tian, Ye Chen, Cheng-Qi Fan, Gui-Hua Tang, Sheng Yin

**Affiliations:** 1School of Pharmaceutical Sciences, Sun Yat-sen University, Guangzhou 510006, China; wenghzh@mail2.sysu.edu.cn (H.-Z.W.); zhujy76@mail2.sysu.edu.cn (J.-Y.Z.); yuanfy@mail2.sysu.edu.cn (F.-Y.Y.); tangzhy23@mail2.sysu.edu.cn (Z.-Y.T.); chenye8@mail.sysu.edu.cn (Y.C.); yinsh2@mail.sysu.edu.cn (S.Y.); 2East China Sea Fisheries Research Institute, Chinese Academy of Fishery Sciences, Shanghai 200090, China; amytian0904@126.com

**Keywords:** diphenyl ether-coupled aromatic bisabolane, aromatic bisabolane dimer, marine-derived fungus, *Aspergillus versicolor*, neuroprotective activity

## Abstract

Chromatographic fractionation of the EtOH extracts of the marine-derived fungus *Aspergillus* *versicolor* A18 has led to the isolation of 11 homo/hetero-dimers of aromatic bisabolane sesquiterpenoids including eight diphenyl ether-coupled aromatic bisabolanes (**1a**/**1b** and **5**–**10**) and three homodimers (**2**–**4**), together with their monomers including three aromatic bisabolanes (**11**–**13**) and two diphenyl ethers (**14** and **15**). Their structures and absolute configurations were elucidated by extensive spectroscopic analysis including HRESIMS, 1D/2D NMR, calculated ECD, and the optical rotatory data. Among the four new compounds, (+/−)-asperbisabol A (**1a**/**1b**), asperbisabol B (**2**), and asperbisabol C (**3**), the enantiomers **1a** and **1b** represent an unprecedented skeleton of diphenyl ether-coupled aromatic bisabolane sesquiterpenoids with a spiroketal core moiety. The neuroprotective effects of selected compounds against sodium nitroprusside (SNP)-induced injury were evaluated in PC12 cells by the MTT assay. Five compounds (**1a**, **6**, and **8**–**10**) showed remarkable neuroprotective activities at 10 μM, being more active than the positive control edaravone.

## 1. Introduction

The genus *Aspergillus* is omnipresent among almost all ecosystems even the circumpolar maritime regions [1]. Previous investigations on *Aspergillus* genus indicated that it was rich in bioactive secondary metabolites with multifarious and intricate structures such as sesquiterpenoids [2,3], meroterpenoids [4,5], alkaloids [6,7], and polyketides [8]. Some of the natural products isolated from *Aspergillus* genus possessed cytotoxic, antimicrobial, acetylcholinesterase inhibitory, and PTP1B enzyme inhibitory activities [2,3,4,5,6,7,8], which have attracted great interest of natural chemists. Bisabolanes, a class of sesquiterpenoids possessing diverse skeletons including aromatic monomers and their home/hetero-dimers, are mainly distributed in plant kingdom, marine-derived animals, and fungi [9]. These bisabolane-type sesquiterpenoids have a broad spectrum of biological properties involving antibacterial, anti-inflammatory, cytotoxic, and antidiabetic activities [9].

Within the past decade, our group has reported a series of compounds with neuroprotective effects, such as polyhydroxypregnane glycosides from *Cynanchum otophyllum* [10], neolignans from *Aristolochia fordiana* [11], mulberry Diels-Alder-type adducts from *Morus alba* [12], and prenylated xanthones from *Garcinia mangostana* [13]. To obtain natural neuroprotective agents, we expanded the coverage of the sources of lead compounds through the in-depth study of marine-derived fungi.

In our ongoing research on novel neuroprotective metabolites from the fungus *Aspergillus versicolor* A18 from South China Sea, a pair of undescribed enantiomeric spiroketal diphenyl ether-coupled aromatic bisabolane sesquiterpenoids (**1a**/**1b**), two new aromatic bisabolane homodimers (**2** and **3**), as well as 12 known analogues (**4**–**15**) (Figure 1), were isolated from the rice media of the strain A18. It is noteworthy that compound **12** was first isolated as a natural product and its absolute configuration was also assigned. The structures of those compounds were determined by comprehensive spectroscopic data, and the absolute configurations were elucidated by ECD calculations or comparing the ECD spectra with those of correlative known analogues. The neuroprotective effects of selected compounds were evaluated. Herein, we report the isolation, structures elucidation, and neuroprotective activities of those compounds.

## 2. Results and Discussion

Asperbisabol A (**1**) was obtained as a colorless oil, whose molecular formula was assigned as C_29_H_34_O_6_ with 13 indices of hydrogen deficiency (IHDs) based on the HRESIMS ion at *m*/*z* 501.2254 [M + Na]^+^ (calcd. for C_29_H_34_O_6_Na 501.2248) and the ^13^C NMR data. The ^1^H NMR data (Table 1) showed signals for three hydroxyl groups [δ_H_ 3.52, 4.78, and 5.51 (each 1H, s)], three singlet methyls [δ_H_ 1.94, 2.10, and 2.23 (each 3H, s)], two doublet methyls [δ_H_ 0.91 (6H, d, *J* = 6.6 Hz)], two olefinic methines [δ_H_ 5.53 (1H, t, *J* = 7.0 Hz) and 6.10 (1H, s)], one 1,2,4-trisubstitutedphenyl [δ_H_ 6.62 (1H, d, *J* = 7.7 Hz), 6.64 (1H, s), and 6.93 (1H, d, *J* = 7.7 Hz)], and an 1,3,4,5-tetrasubstitutedphenyl [δ_H_ 6.26 (1H, s) and 6.33 (1H, s)]. The ^13^C NMR data of compound **1** (Table 1), in conjunction with DEPT and HSQC spectra, displayed the presence of 29 carbon resonances attributable to one conjugated ketocarbonyl carbon, nine sp^2^ quaternary carbons including four oxygenated ones, two oxygenated sp^3^ quaternary carbons, seven sp^2^ methine, one sp^3^ methine, four sp^3^ methylene, and five sp^3^ methyl. It was obvious from the comparison of the information with the data of the known diphenyl ether-coupled aromatic bisabolane sesquiterpenoid expansol D (**7**) [14] and from analysis of the 2D NMR data of **1** that asperbisabol A was a heterodimer comprising an aromatic bisabolane unit and a highly variable diphenyl ether moiety, parts a and b, as shown in Figure 2.

The aromatic bisabolane unit (part a) was defined by the analysis of ^1^H-^1^H COSY, HMBC, and NOESY correlations. The key HMBC correlations from H_3_-14 to C-1, C-7, and C-8, from H_2_-15 to C-3, C-4, and C-5, and from 2-OH to C-1, C-2, and C-3 easily allowed the partial structure (C-8–C-12/13) established by the ^1^H−^1^H COSY correlations of H-8/H_2_-9/H_2_-10/H-11/H_3_-12(H_3_-13), the 1,2,4-trisubstitutedphenyl ring, the olefinic carbon (C-7), the methyl (CH_3_-14), and the methylene (CH_2_-15) to form an aromatic bisabolane unit. In addition, the observed NOESY correlation of H-9*β* and H_3_-14 defined its Δ^7^ double bond was a *trans*-configuration (Figure 3).

The structure of highly variable diphenyl ether moiety (part b) was deduced by the interpretation of HMBC, ^13^C NMR, and IHDs data. The HMBC correlations from H_3_-14′ to C-8′, C-9′, and C-13′, from H-9′ to C-10′ and C-11′, and from H-13′ to C-11′ and C-12′ confirmed that the 1,3,4,5-tetrasubstitutedphenyl ring (ring A) was a 5-methylbenzene-1,2,3-triol unit. In the HMBC spectrum, the observed correlations from 4′-OH to C-3′, C-4′, and C-5′, from H_3_-7′ to C-1′, C-2′, and C-6′, from H-2′ to C-4′, and from H-6′to C-5′ constructed ring B featured with a *α*,*β*-unsaturated ketone. In view of 12 of the 13 IHDs accounted for by two phenyls, a six-membered ring, two double bonds, and a ketocarbonyl group, as well as the chemical shift of C-5′ (δ_C_ 118.9), C-11′ (δ_C_ 132.2), and C-12′ (δ_C_ 138.1), we deduced that a five-membered ring (ring C) connected rings A and B to build a spiroketal core moiety. Thus, part b of **1** was a spiroketal skeleton derived from a diphenyl ether. The HMBC correlations from H_2_-15 to C-3′, C-4′, and C-5′ and from 4′-OH to C-15 established that the aromatic bisabolane unit (part a) and highly variable diphenyl ether moiety (part b) were linked by the C-15–C-4′ bond. Finally, the planar structure of asperbisabol A (**1**) was elucidated.

In the NOESY spectrum of **1**, the observed NOESY correlations of H-15*β*/H-6′*β* and 2-OH/10′-OH suggested that the relative configurations of C-4′ and C-5′ were defined as shown in Figure 3. The optical rotatory data of **1** was zero, which indicated that asperbisabol A (**1**) may be a racemate. Subsequent chiral resolution of **1** by semipreparative HPLC afforded the corresponding enantiomers **1a** (${\left[\alpha \right]}_{D}^{20}$ = +37.00) and **1b** (${\left[\alpha \right]}_{D}^{20}$ = −26.00). The absolute configurations of this pair of enantiomers were determined by comparison of their experimental ECD spectra with the calculated ECD spectra of (4′*R*,5′*R*)-**1** and (4‘*S*,5′*S*)-**1**. As shown in Figure 4, the experimental ECD curves of **1a** and **1b** matched well with the calculated ECD spectra of (4′*R*,5′*R*)-**1** and (4‘*S*,5′*S*)-**1****,** respectively. Therefore, the absolute configurations of **1a** (4′*R*,5′*R*) and **1b** (4‘*S*,5′*S*) were unambiguously determined, and given their trivial names (+)-asperbisabol A and (−)-asperbisabol A, respectively.

Compound **2** was obtained as a colorless oil. The molecular formula of **2** was settled as C_31_H_46_O_6_ with 9 IHDs by the HRESIMS ion at *m*/*z* 537.3210 [M + Na]^+^ (calcd. for C_31_H_46_O_6_Na 537.3187). The ^1^H NMR data (Table 1) displayed the characteristic signals assigned to seven methyls (including one oxygenated), two 1,2,4-trisubstitutedphenyls, and two hydroxyl groups. The ^13^C NMR data (Table 1) in combination with DEPT and HSQC spectra confirmed seven sp^2^ quaternary carbons (including one carbonyl and two other oxygenated ones), two oxygenated sp^3^ quaternary carbons, six sp^2^ methines, two sp^3^ methines, seven sp^3^ methylenes (including an oxygenated), and seven sp^3^ methyls (including a methoxyl). The ^1^H NMR and ^13^C NMR data of **2** closely resembled those of peniciaculin B (**4**), except for the presence of an additional methoxyl (δ_H_ 3.13, δ_C_ 50.4). Key HMBC correlation from 7-OMe to C-7 confirmed the position of the methoyl group. The ECD spectrum of compound **2** was similar to that of **4** in terms of positive Cotton effect (CE) from 260 nm to 320 nm and negative CE around 210 nm (Appendix A) [3]. Therefore, the absolute configuration of **2** was also logically established as 7*S*,7′*S* and given the trivial name asperbisabol B.

Compound **3** was isolated as a colorless oil and assigned a positive HRESIMS ion at *m*/*z* 521.2689 [M + K]^+^ (calcd. for C_30_H_42_O_5_K 521.2664), which well matched a molecular formula of C_30_H_42_O_5_ with 10 IHDs. Through cumulative analysis of the ^1^H NMR and ^13^C NMR data of compounds **3** and **4**, it made sense that **3** was a dehydration product of **4**. This speculation was supported by the presence of a double bond (δ_H_ 5.56; δ_C_ 131.6 and 132.3) in **3** rather than an oxygenated sp^3^ quaternary carbon (δ_C_ 78.9) and a sp^3^ methylene (δ_H_ 1.80 and 1.89; δ_C_ 43.0) in **4**. The HMBC correlations from H_3_-14 to C-1, C-7, and C-8 further confirmed the location of the double bond. The positive CE from 260 nm to 320 nm and negative CE around 210 nm (Appendix A) [3] unequivocally established the absolute configuration of **3** as 7′*S* and given the trivial name asperbisabol C.

In addition to the above four new homo/hetero-dimers of aromatic bisabolanes (**1a**, **1b**, **2**, and **3**), 12 previously described compounds **4**–**15** were isolated. They were identified as peniciaculin B (**4**) [3], expansol E (**5**) [14], expansol C (**6**) [14], expansol D (**7**) [14], expansol A (**8**) [15], aspertenol A (**9**) [16], peniciaculin A (**10**) [3], (*Z*)-5-(hydroxymethyl)-2-(6′-methylhept-2′-en-2′-yl)phenol (**11**) [17], (*R*)-3-hydroxy-4-(2-hydroxy-6-methylheptan-2-yl)benzaldehyde (**12**) [18], aspergillusene E (**13**) [19], 3-(3-methoxy-5-methylphenoxy)-5-methylphenol (3-*O*-methyldiorcinol, **14**) [20,21], and 3,3′-dihydroxy-5,5′-dimethyldiphenyl ether (diorcinol, **15**) [20,21] by analyzing their NMR data and comparing with those reported in the correlative papers. Among them, **12** was first isolated as a natural product and its absolute configuration was assigned as *R* by the antipodal CE comparing with **2**–**4** (Appendix A).

The neuroprotective effects of **1a**, **1b**, and **2**–**14** against sodium nitrosprusside (SNP, 700 μM) induced injury were evaluated by the MTT assay in PC12 cells. The results of preliminary screening at a concentration of 10 μM (Figure 5A) showed that compounds **1a**, **6**, and **8**–**10** exhibited more neuroprotective activities than that of the positive control edaravone (Eda, a free radical scavenger). Furthermore, these five active compounds have potent effects in a concentration-dependent manner in the range of 2.5–10 μM (Figure 5B).

## 3. Materials and Methods

### 3.1. General Experimental Procedures

Optical rotations were measured on an Anton Paar MCP200 polarimeter (Graz, Austria). Circular dichroism spectra and UV spectra were obtained on an Applied Photophysics Chirascan spectrometer (Surrey, UK). HRESIMS were performed on a Shimadzu LCMS-IT-TOF spectrometer (Kyoto, Japan). NMR spectra were measured on Bruker Ascend TM 500 (Bremerhaven, Germany) and Bruker Avance III 400 (Zurich, Switzerland) spectrometer at 25 °C with TMS as the internal standard. Silica gel (100–200, 200–300 and 300–400 mesh, Qingdao Haiyang Chemical Co., Ltd., Qingdao, China), D101 macroporous resin (Donghong Chemical Co., Ltd., Changzhou, China), ODS reversed-phase silica gel (12 nm, S-50 μm, YMC Co., Ltd., Komatsu, Ishikawa, Japan), and Sephadex LH-20 gel (Amersham Biosciences, Shanghai, China) were used for column chromatography (CC). Semi-preparative HPLC was performed with a YMC-pack ODS-A column (10 × 250 mm, S-5 μm), a NanoChrom ChromCoreTM 5-120 C18 column (250 × 10 mm, 5 μm), or a Phenomenex Lux cellulose-2 chiral column (10 × 250 mm, 5 μm, 12 nm) under Shimadzu LC-20 AT equipped with a SPD-M20A PDA detector (Kyoto, Japan). Almost all chemical solvents were of analytical grade (Guangzhou Chemical Reagents Company, Ltd., Guangzhou, China) while acetonitrile (MeCN) was of HPLC grade (Grace Chemical Technology Co., Ltd., Qingdao, China).

### 3.2. Fungal Material

The fungal strain *Aspergillus versicolor* A18 was isolated from a surface water sample collected in South China Sea and identified as *Aspergillus versicolor* on the base of the ITS region (GenBank MT5827511) [22]. The voucher specimen is deposited in East China Sea Fisheries Research Institute.

### 3.3. Fermentation and Extraction

The strain *A*. *versicolor* A18 was cultured on PDA plates (PDA media 24.0 g, agar 18.0 g and sea salt 30.0 g in 1.0 L H_2_O) at 28 °C for 7 days. The seed medium (PDB media 24.0 g and sea salt 30.0 g in 1.0 L H_2_O) was inoculated with strain *A*. *versicolor* A18 and incubated at 28.0 °C for 3 days on a rotating shaker (180 rpm). For chemical investigations, a large-scale fermentation of *A*. *versicolor* A18 was incubated for 28 days at 28 °C in 1.5 L × 40 conical flasks (each flask contained 450.0 g rice and 300.0 mL H_2_O with 3% salinity). After incubating, every flask was ultrasonically extracted with 4 × 0.4 L 95% EtOH for 30 min. The combined extract was subjected to nanofiltration membrane (300 D) for desalination and concentration. Then total 30 L concentrated solution was evaporated under reduced pressure to yield a dark brown gum, which was redissolved in 2 L water and subsequently extracted three times with petroleum ether (PE, 3 L each time) and five times with EtOAc (3 L each time) to afford PE fraction and EtOAc fraction.

### 3.4. Isolation and Purification

The EtOAc fraction (62 g) was subjected to macroporous resin column (MeOH/H_2_O, 3/7→10/0, *v*/*v*) to afford four fractions (Frs. A–D). Compound **15** (8.7 mg) was purified by semipreparative HPLC (50% MeCN/H_2_O, YMC-pack ODS-A column, 3 mL/min, t*_R_* 13.8 min) from Fr. C. Fr. D (16 g) was further subjected to CC on silica gel using varying polarities of PE/EtOAc (1/0→0/1, *v*/*v*) to afford six subfractions (Frs. D1–D6).

Fr. D5 (1.0 g) was separated by Sephadex LH-20 CC (CH_2_Cl_2_/MeOH, 1/1, *v*/*v*) and followed by silica gel (CH_2_Cl_2_/MeOH, 300/1, *v*/*v*) CC to obtain several sub-fractions. From the sub-fraction Fr. D5C, compound **1** was purified by semipreparative HPLC (85% MeCN/H_2_O, C18 column, 3 mL/min, t*_R_* 12.1 min). Subsequently, the chiral resolution of **1** by semipreparative HPLC (60% MeCN/H_2_O, chiral column, 3 mL/min) yielded (−)-**1** (3.9 mg, t*_R_* 13.1 min) and (+)-**1** (2.8 mg, t*_R_* 13.9 min).

Fr. D4 (520 mg) was divided into three fractions (Frs. D4A–D4C) by Sephadex LH-20 (CH_2_Cl_2_/MeOH, 1/1, *v*/*v*), and Fr. D4C (117 mg) was further purified by CC on silica gel (PE/EtOAc, 10/1, *v*/*v*), ODS reversed-phase silica gel (MeOH/H_2_O, 5/5→3/7, *v*/*v*), and semipreparative HPLC (80% MeCN/H_2_O, YMC-pack ODS-A column, 3 mL/min) to obtain **6** (3.7 mg, t*_R_* 14.1 min) and **7** (6.2 mg, t*_R_* 15.6 min).

Fr. D3 (1.5 g) was separated by ODS reversed-phase silica gel (MeOH/H_2_O, 5/5→10/0, *v*/*v*) to give seven sub-fractions (Frs. D3A–D3G). Fr. D3C (45 mg) was chromatographed over semipreparative HPLC (65% MeCN/H_2_O, YMC-pack ODS-A column, 3 mL/min) to obtain **11** (1.1 mg, t*_R_* 12.4 min), **14** (3.1 mg, t*_R_* 13.2 min), and **12** (1.2 mg, t*_R_* 14.2 min). Fr. D3D (45 mg) was subjected to silica gel CC (PE/EtOAc, 20/1→10/1, *v*/*v*) and further purified by semipreparative HPLC (65% MeCN/H_2_O, YMC-pack ODS-A column, 3 mL/min) to give **13** (1.1 mg, t*_R_* 20.4 min). Compound **5** was obtained from Fr. D3E (234 mg) by silica gel CC (PE/acetone, 20/1, *v*/*v*) and semipreparative HPLC (70% MeCN/H_2_O, YMC-pack ODS-A column, 3 mL/min, 3.6 mg, t*_R_* 22.8 min). Fr. D3F (351 mg) was separated by Sephadex LH-20 (CH_2_Cl_2_/MeOH, 1/1, *v*/*v*) and silica gel CC was eluted with a gradient solvent system of PE/EtOAc (from 100:0 to 0:100) to yield five sub-fractions (Frs. D3F1–D3F5). Fr. D3F2 (69.1 mg) was further purified by semipreparative HPLC (78% MeCN/H_2_O, chiral column, 3 mL/min) to obtain **2** (4.8 mg, t*_R_* 16.7 min). Compound **3** (2.3 mg, t*_R_* 15.9 min) was obtained from Fr. D3F3 (90.1 mg) by semipreparative HPLC (90% MeCN/H_2_O, C18 column, 3 mL/min). Fr. D3F4 (72.0 mg) was subjected to semipreparative HPLC (90% MeCN/H_2_O, YMC-pack ODS-A column, 3 mL/min) to obtain **10** (5.1 mg, t*_R_* 14.8 min), **9** (2.5 mg, t*_R_* 16.5 min), **8** (3.5 mg, t*_R_* 19.5 min), and **4** (15.3 mg, t*_R_* 20.9 min).

(+)-Asperbisabol A (**1a**): colorless oil; ${\left[\alpha \right]}_{D}^{20}$ = +37.00 (c 0.1, MeCN); UV (MeCN): λ_max_ (logε) 209 (1.61), 283 (0.16) nm; ECD (MeCN): λ_ext_ (Δε) 197 (−4.47), 210 (−14.34), 236 (+6.77), 323 (+3.29) nm; HRESIMS at *m*/*z* 501.2254 [M + Na]^+^ (calcd. for C_29_H_34_O_6_Na, 501.2248) and *m*/*z* 477.2279 [M − H]^−^ (calcd. for C_29_H_33_O_6_, 477.2283); ^1^H and ^13^C NMR data, see Table 1.

(−)-Asperbisabol A (**1b**): colorless oil; ${\left[\alpha \right]}_{D}^{20}$ = −26.00 (c 0.1, MeCN); ECD (MeCN): λ_ext_ (Δε) 197 (+3.11), 209 (+10.82), 236 (−4.74), 323 (−2.34) nm; UV, NMR, and HRESIMS are the same as those of **1a**.

Asperbisabol B (**2**): colorless oil; UV (MeCN): ${\left[\alpha \right]}_{D}^{20}$ = +3.00 (c 0.1, MeCN); λ_max_ (logε) 195 (3.88), 214 (2.62), 245 (0.69), 286 (0.32) nm; ECD (MeCN): λ_ext_ (Δε) 192 (+9.21), 213 (−5.55), 297 (+1.75) nm; HRESIMS at *m*/*z* 537.3210 [M + Na]^+^ (calcd. for C_31_H_46_O_6_Na, 537.3187); ^1^H and ^13^C NMR data, see Table 1.

Asperbisabol C (**3**): colorless oil; ${\left[\alpha \right]}_{D}^{20}$ = +4.00 (c 0.3, MeCN); UV (MeCN): λ_max_ (logε) 224 (3.20), 246 (3.34), 288 (1.48) nm; ECD (MeCN): λ_ext_ (Δε) 197 (+0.28), 213 (−2.32), 239 (+0.06), 301 (+0.91) nm; HRESIMS at *m*/*z* 521.2689 [M + K]^+^ (calcd. for C_30_H_42_O_5_K, 521.2664); ^1^H and ^13^C NMR data, see Table 1.

(*R*)-3-Hydroxy-4-(2-hydroxy-6-methylheptan-2-yl)benzaldehyde (**12**): colorless oil; UV (MeCN): λ_max_ (logε) 195 (0.92), 222 (0.65), 254 (0.43) nm; ECD (MeCN): λ_max_ (Δε) 194 (−5.26), 212 (+7.89), 240 (−1.44), 256 (+0.71), 282 (−2.72) nm; ^1^H and ^13^C NMR data, see Appendix A.

### 3.5. ECD Calculation for Assigning the Absolute Configurations of ***1a*** and ***1b***

The absolute configurations of **1a** and **1b** were determined by quantum chemical calculations of their theoretical ECD spectra. (4′*R*,5′*R*)-**1**, one of the enantiomers for **1**, was arbitrarily chosen for theoretical studies. Conformational analyses were first carried out via Monte Carlo searching using molecular mechanism with MMFF force field in the *Spartan 18* program. The results showed 20 lowest energy conformers for **1** within an energy window of 2.0 Kcal/mol. These conformers were reoptimized using DFT at the B3LYP/6-31G(d) level in gas phase using the *Gaussian 09* program. 11 conformers of **1** (Appendix A) with the relative Gibbs free energies (ΔG) in the range of 0–1.5 Kcal/mol were refined and considered for next step. All the reoptimized conformers were applied for theoretical ECD calculation. The energies, oscillator strengths, and rotational strengths of the first 30 electronic excitations were calculated using the TD-DFT methodology at the M062X/TZVP level in PCM (acetonitrile). The ECD spectra were simulated by the overlapping Gaussian function (σ = 0.40 eV), in which velocity rotatory strengths of the first 18 exited states for **1** were adopted. To get the final ECD spectrum of each compound, the simulated spectra of the lowest energy conformers were averaged according to the Boltzmann distribution theory and their relative Gibbs free energy (ΔG). The theoretical ECD curve of (4′*S*,5′*S*)-**1** was obtained by directly reversing that of (4′*R*,5′*R*)-**1**.

### 3.6. Neuroprotective Bioassays

Compounds **1a**, **1b**, and **2**–**14** as well as the positive control edaravone (Aladdin, Shanghai, China) were dissolved in DMSO (Sigma-Aldrich, Shanghai, China) as a stock, and the tested compounds was further diluted by DMEM medium (Gibco, Beijing, China) into three gradient concentrations (2.5, 5, and 10 μM). PC12 cells were digested and seeded into 96-well plates at a density of 5 ×10^3^ cells per well and cultured in DMEM medium with 5% CO_2_ for 24 h. Then the cell culture medium was replaced by DMEM medium containing different concentrations of compounds for pretreatment for 2 h and then treated with 700 μM SNP (Sigma-Aldrich, Shanghai, China) for another 24 h. About 10 µL of MTT (Beyotime Institute of Biotechnology, Shanghai, China) (5 mg/mL) was added into each well and incubated at 37 °C for 3.5 h. Afterwards, the supernatant was removed and the crystals were dissolved in 100 µL DMSO. The optical absorbance at 570 nm was read with an EPOCH 2 microplate reader (BioTek Devices, San Mateo, CA, USA). The experiments were repeated three times.

## 4. Conclusions

In conclusion, a total of 16 natural products including four new ones (**1a**, **1b**, **2**, and **3**) were isolated from the marine-derived fungus *Aspergillus versicolor* A18. Their structures were identified as diphenyl ether-coupled aromatic bisabolanes (**1a**/**1b** and **5**–**10**), homodimers of aromatic bisabolanes (**2**–**4**), aromatic bisabolanes (**11**–**13**), and diphenyl ethers (**14** and **15**). The pair of enantiomeric diphenyl ether-coupled aromatic bisabolanes, (+/−)-asperbisabol A (**1a**/**1b**), represented a rare heterodimers characteristic of a spiroketal core moiety, which expanded the structural diversity of this type of bisabolane sesquiterpenoids. Compounds **1a**, **6**, and **8**–**10** showed more potent neuroprotective activity than that of the positive control edaravone, which shed light on the bioactivity evaluation of aromatic bisabolanes.

## Figures and Tables

**Figure 1 marinedrugs-20-00322-f001:**
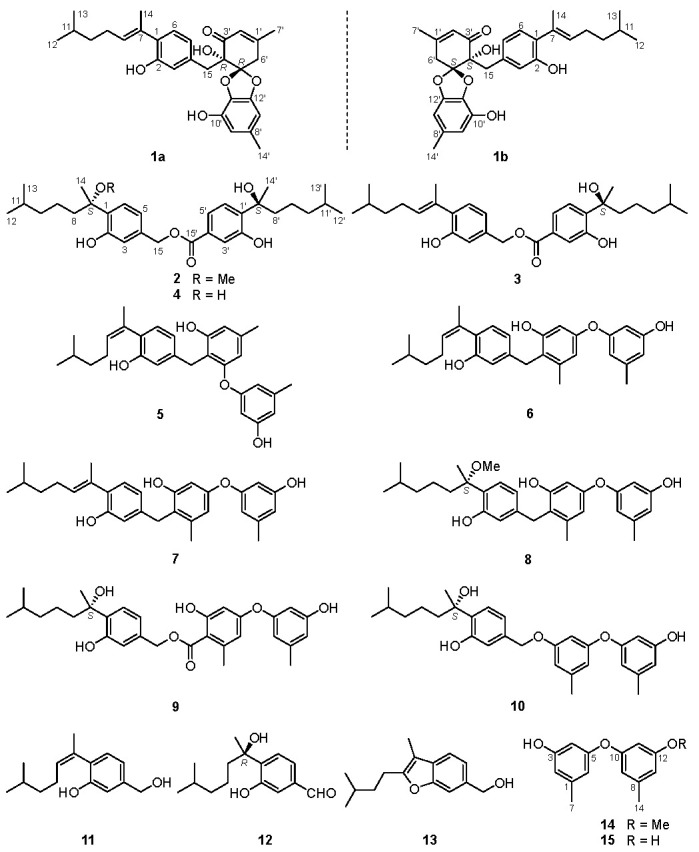
Chemical structures of compounds **1**–**15**.

**Figure 2 marinedrugs-20-00322-f002:**
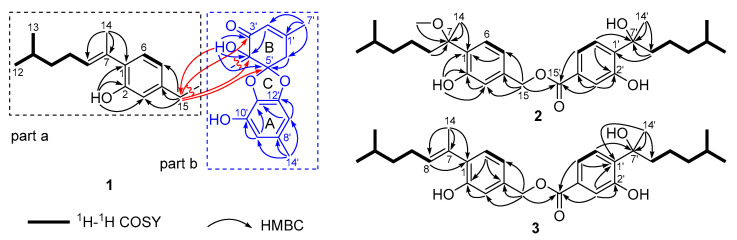
Key ^1^H–^1^H COSY and HMBC correlations of **1**–**3**.

**Figure 3 marinedrugs-20-00322-f003:**
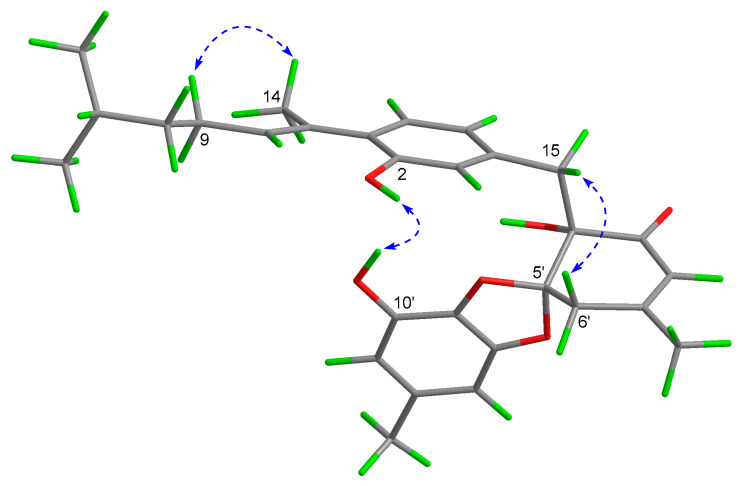
Key NOESY correlations of **1**.

**Figure 4 marinedrugs-20-00322-f004:**
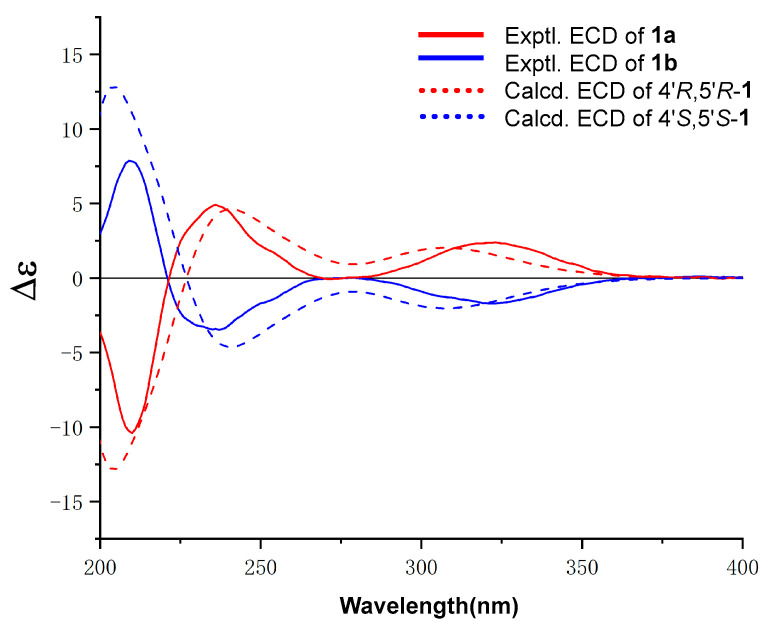
Experimental ECD spectra of **1a** (red solid line) and **1b** (blue solid line) and B3LYP/6-31G(d) calculated ECD spectra of (4′*R*,5′*R*)-**1** (red dash line) and (4′*S*,5′*S*)-**1** (blue dash line).

**Figure 5 marinedrugs-20-00322-f005:**
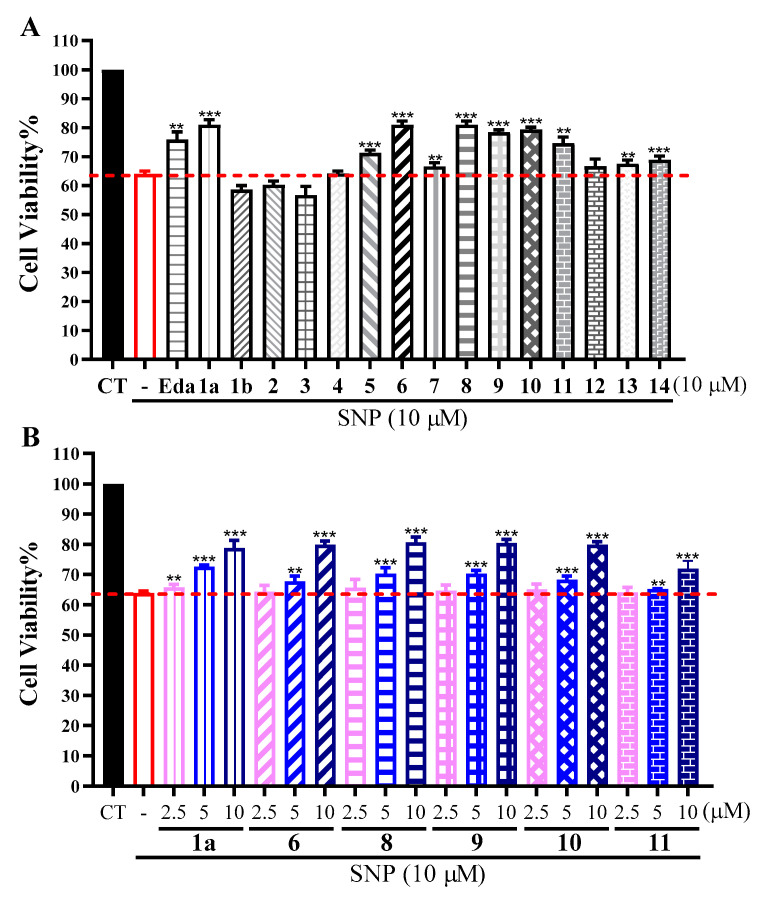
Neuroprotective activity assay: (**A**) Effects of compounds **1a**, **1b**, and **2**–**14** (10 μM) on SNP-induced neurotoxicity in PC12 cells; (**B**) effects of active compounds **1a**, **6**, and **8**–**11** on SNP-induced neurotoxicity in PC12 cells in the range of 2.5–10 μM. *** *p* < 0.01 compared with the SNP group; ** *p* < 0.05 compared with the SNP group.

**Table 1 marinedrugs-20-00322-t001:** ^1^H NMR and ^13^C NMR data for compounds **1**–**3** in CDCl_3_ (δ in ppm).

Position	1 ^a^		2 ^b^		3 ^a^	
	δ_C_, Type	δ_H_ (*J* in Hz)	δ_C_, Type	δ_H_ (*J* in Hz)	δ_C_, Type	δ_H_ (*J* in Hz)
1	129.9, C		128.0, C		131.1, C	
2	151.6, C		156.1, C		152.2, C	
3	117.4, CH	6.64, s	116.4, CH	6.92, s	115.0, CH	6.98, s
4	134.3, C		137.3, C		136.2, C	
5	122.4, CH	6.62, d (7.7)	119.1, CH	6.89, d (7.9)	119.9, CH	6.93, d (7.8)
6	127.9, CH	6.93, d (7.7)	127.7, CH	7.00, d (7.9)	128.6, CH	7.07, d (7.8)
7	131.7, C		82.9, C		131.6, C	
8	132.0, CH	5.53, t (7.0)	40.2, CH_2_	1.82, m 1.93, m	132.3, CH	5.56, t (7.1)
9	26.5, CH_2_	2.20, q (8.0)	21.8, CH_2_	1.33, m	26.5, CH_2_	2.22, q (7.5)
10	38.7, CH_2_	1.32, q (7.5)	39.1, CH_2_	1.14, m	38.7, CH_2_	1.33, m
11	27.9, CH	1.60, m	27.9, CH	1.50, m	27.9, CH	1.60, m
12	22.6, CH_3_	0.91, d (6.6)	22.7, CH_3_	0.83, d (6.6)	22.7, CH_3_	0.93, d (6.6)
13	22.6, CH_3_	0.91, d (6.6)	22.8, CH_3_	0.84, d (6.6)	22.7, CH_3_	0.93, d (6.6)
14	18.0, CH_3_	1.94, s	22.4, CH_3_	1.58, s	18.0, CH_3_	1.98, s
15	40.3, CH_2_	2.94, d (13.4) 3.31, d (13.4)	66.2, CH_2_	5.27, s	66.4, CH_2_	5.27, s
1′	157.1, C		134.5, C		134.5, C	
2′	124.0, CH	6.10, s	156.3, C		156.3, C	
3′	197.6, C		119.2, CH	7.56, s	119.0, CH	7.55, s
4′	81.8, C		130.7, C		130.7, C	
5′	118.9, C		120.8, CH	7.52, d (8.2)	120.8, CH	7.52, d (8.2)
6′	42.0, CH_2_	3.01, d (19.1) 3.14, d (19.0)	126.4, CH	7.05, d (8.2)	126.4, CH	7.04, d (8.1)
7′	24.3, CH_3_	2.10, s	79.2, C		79.2, C	
8′	132.2, C		43.1, CH_2_	1.81, m 1.90, m	43.1, CH_2_	1.79, m 1.91, m
9′	102.4, CH	6.33, s	21.8, CH_2_	1.29, m	21.8, CH_2_	1.28, m
10′	148.6, C		39.3, CH_2_	1.14, m	39.1, CH_2_	1.14, m
11′	132.2, C		27.9, CH	1.50, m	27.9, CH	1.49, m
12′	138.1, C		22.7, CH_3_	0.83, d (6.6)	22.7, CH_3_	0.82, d (6.6)
13′	111.3, CH	6.26, s	22.7, CH_3_	0.83, d (6.6)	22.7, CH_3_	0.83, d (6.6)
14′	21.5, CH_3_	2.23, s	29.3, CH_3_	1.66, s	29.3, CH_3_	1.66, s
15′			166.3, C		166.3, C	
2-OH		5.51, s		8.88, s		
2′-OH				9.30, s		9.30, s
4′-OH		3.52, s				
10′-OH		4.78, s				
7-OMe			50.6, CH_3_	3.22, s		

^a^ Measured at 500 MHz. ^b^ Measured at 400 MHz.

## Data Availability

Not applicable.

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
