# Peer review of "Homo/Hetero-Dimers of Aromatic Bisabolane Sesquiterpenoids with Neuroprotective Activity from the Fungus Aspergillus versicolor A18 from South China Sea"

_marinedrugs, 2022, doi:10.3390/md20050322_

Round 1
Reviewer 1 Report
Dear Authors,I would like to express my praise for such an interesting
description of the establishment of a mixture of enantiomers and their
further separation. Ð owever, I ask you to draw your attention to some
controversial points that are not fundamental.
Line 53. "It is noteworthy that (R)-3-hydroxy-4-(2-hydroxy-6-methylheptan-2-yl)benzaldehyde (12)
was firstly isolated as a natural product and its absolute configuration was also assigned."
Determination the absolute configuration of
compound 12 is not discussed in any way either
in the article or in the supplementary. Moreover, supplementary Figure S58,
only experimental data are given without discussion,
for compounds 4 and 12 (figure S58)
for positive and negative Cotton Effects shold be to give λmax (∆ε) too
(like in experimrntal part for compounds 1a, 1b, 2 and 3)..
Line 19, Line 104, the phrases "were elucidated by synergetic
utilization of extensive spectroscopic analysis" and
"the gloss planar structure" do not look purely chemical, but i'm not native speaker.
The trivial name of compounds diorcinol (15) and 3-O-methyldiorcinol (14)
simpler and more common in the scientific community. Might be worth
mentioning them too on page 6 line 156?
For example https://www.mdpi.com/1660-3397/20/1/67
Reviewer 2 Report
The manuscript is well-formed, the text needs some correction. Necessary changes are indicated in the text of the manuscript.

Reviewer 3 Report
This manuscript describes the isolation and structure elucidation of four new natural products, (±)-asperbisabol A (1a/b), asperbisabol B (2) and asperbisabol C (3), along with twelve known natural products, peniciaculins A (10) and B (4), espansols A (8), E (5), C (6) D (7), asperenol A (9), (Z)-5-(hydroxymethyl)-2-(6'-methylhept-2'-en-2'-yl)phenol (11), (R)-3-hydroxy-4-(2-hydroxy-6-methylheptan-2-yl)benzaldehyde (12), aspergillusene E (13), 3-(3-methoxy-5-methylphenoxy)5-methylphenol (14), and 3,3'-dihydroxy-5,5'-dimethyldiphenyl ether (15), from a cultivation of the marine-derived fungus Aspergillus versicolor A18.
New Natural Products
Structures assigned to the new natural products 1a/b, 2 and 3 are based on detailed spectroscopic analysis. As the structure elucidations rely heavily on correct 1D and 2D NMR assignments and correlations it is remiss of the authors to not provide a full tabulation of the 1D and 2D NMR data for each of the new compounds. While the inclusion of (unannotated) images of NMR spectra in the Supporting Information can be informative, and there is no doubt that selected 2D NMR correlations can provide very informative "arrow" diagrams (Fig 2), these are not a substitute for comprehensive documentation and analysis. Given the potential for large numbers of 2D NMR correlations, it's important to know that all the correlations fit the proposed structure, and not simply focus on those cherry-picked for the arrow diagrams. Likewise, with 1D NMR it is unwise to make assignments to overlapping resonances (protons or carbons) – of which there are many in Table 1. Instead, when tabulating NMR data, it is prudent to use superscripts to flag overlapping resonances and provide suitable footnotes, indicating where unique assignments cannot be made. This is especially important as, in many cases the assignment of individual 1D NMR resonances relies on diagnostic 2D NMR correlations, and 2D NMR correlations in turn rely on correct 1D NMR assignments. Significantly, where 1D NMR resonances overlap this impacts the ability to assign and interpret 2D NMR correlations, which in turn constrains the practical use of assignments/correlations associated with overlapping resonances.
The authors provide no comprehensive tabulation of the NMR data for the new compounds, and Table 1 includes numerous "unique" assignments based on overlapping resonances, which in turn inform "unique" 2D NMR correlations in Figure 2. Indeed, the authors do not acknowledge any overlapping NMR resonances when even a cursory scan of the NMR data reveals this is clearly not the case. These overlapping issues need to be resolved, and the NMR based structure elucidation adjusted accordingly.
Furthermore, the authors only provide NMR data in the manuscript (ie Table 1) and the SI for the racemic mixture 1, and not for the purported enantiomers 1a and 1b? Why? The ability to compare these two separate NMR data sets is critical to validating the case that 1a and 1b are enantiomers. Without such a comparison it's not possible to exclude the option that 1a and 1b are either C-4' or C-5' diastereomers (which might be expected to have very similar, but not necessarily identical NMR spectra). For the authors to substantiate the claim to have isolated and identified the enantiomers 1a and 1b they need to (at the very least) characterise and compare their 1H NMR data.
Known Co-metabolites
Where the authors claim to have isolated known natural products (ie 4-15) it is imperative that sufficient spectroscopic evidence be provided (even if only in the Supporting Information) to validate the purity and assigned structures. This includes acquiring and tabulating NMR data, and making appropriate comparisons to literature data (in the same solvent). Where known compounds are chiral it is also essential that specific rotations be measured, and compared to appropriate literature reports (in the same solvent).
The only data provided to validate the structures assigned to 4-15 are unannotated 1H and 13C NMR spectra in the SI. None of the NMR data is tabulated and no directed comparisons are disclosed with literature data in the same solvent, and no HRMS measurements are reported to confirm molecular formulae. Likewise, although 3-4, 7-10 and 12 are chiral, no optical rotation measurements are disclosed, and no comparisons made to the literature (again in the same solvent). This is particularly remiss given that the authors make the claim that 12 is antipodal to the co-metabolites 2-4 and 8-10.
Unambiguous proof of the structures of known compounds is particularly important where claims are made with respect to their biological properties. It is not sufficient to just state that known compounds were isolated and had the same spectroscopic properties as in the literature.
Bioactivity
The neuroprotective assay is far from compelling, and claims to a potent neuroprotective effect in the 5-10 uM range seem exaggerated. Moreover, phenols are well known for exhibiting antioxidant properties, and the authors have not excluded the possibility that the putative neuroprotective effect is merely an antioxidant mediated depletion of NO superoxide.
In summary, the manuscript does report some new natural products. However, the NMR data relied on the support structure assignments is not well documented, and is compromised by over-interpretation of overlapping resonances, making it very difficult to confirm that the propose structures are correct. Although claiming two enantiomers 1a and 1b, the authors do not provide NMR analysis of these enantiomers, instead limiting the NMR analysis to the mixed NMR data for 1a/b. While the authors claim to have assigned structures to the known compounds 4-15, in fact they have merely stated not proved these assignments. Significantly, chiral known compounds are not appropriately characterised so any claims to absolute chemistry is unsubstantiated. The neuroprotective assay is naïve and claims to potency exaggerated and potentially misplaced.
The manuscript could be suitable for publication but only subject to the authors "effectively" addressing the issues outlined above.
Round 2
Reviewer 3 Report
The authors have largely provided the additional data/clarity as requested - which is fine. The only concern I still have is that they state they have added optical rotations or ECD for all known compounds to the SI, however, I could not find this data for the three known compounds 8-10? Did I miss something? Where is this data, and if its not provided how can the authors assert that they know the absolute configuration, particularly given the antipodal co-metabolite 12? Assuming the authors either add the data or remove the claim regarding assignment of absolute configurations to 8–10, the manuscript is suitable for publication.
